# In vivo targeted and deterministic single-cell malignant transformation

Pierluigi Scerbo[1,2]*, Benjamin Tisserand[1], Marine Delagrange[1,3], Héloise Debare[1], David Bensimon[1,4]*, Bertrand Ducos[1,3]*

[1]Laboratoire de Physique de l'Ecole Normale Supérieure LPENS, ENS, PSL Research University, CNRS, Sorbonne Université, Université de Paris, Paris, France; [2]Inovarion, Paris, France; [3]High Throughput qPCR Core Facility of the ENS, Ecole Normale Supérieure, PSL Research University, IBENS, Paris, France; [4]Dept. Chemistry and Biochemistry, UCLA, Los Angeles, United States

## eLife Assessment

This **important** study employs an optogenetics approach aimed at activating oncogene (KRASG12V) expression in a single somatic cell, with a focus on following the progression of activated cell to examine tumourigenesis probabilities under altered tissue environments. Although the description of the methodologies applied is **incomplete**, the authors propose a mechanism whereby reactivation of re-programming factors correlates with the increased likelihood of a mutant cell undergoing malignant transformation. This work will be of interest to developmental and cancer biologists, especially in relation to the genetic tools described.

## Abstract

Why does a normal cell possibly harboring genetic mutations in oncogene or tumor suppressor genes becomes malignant and develops a tumor is a subject of intense debate. Various theories have been proposed but their experimental test has been hampered by the unpredictable and improbable malignant transformation of single cells. Here, using an optogenetic approach we permanently turn on an oncogene (KRASG12V) in a single cell of a zebrafish brain that, only in synergy with the transient co-activation of a reprogramming factor (VENTX/NANOG/OCT4), undergoes a deterministic malignant transition and robustly and reproducibly develops within 6 days into a full-blown tumor. The controlled way in which a single cell can thus be manipulated to give rise to cancer lends support to the 'ground state theory of cancer initiation' through 'short-range dispersal' of the first malignant cells preceding tumor growth.

## Introduction

How cancer arises from a single normal cell is still the subject of active debate, affecting intervention strategies. While many cells may harbor oncogenic mutations, only a few unpredictably end up developing a full-blown tumor (*Cairns, 1975*; *Riva et al., 2020*; *Adashek et al., 2020*; *Hanahan and Weinberg, 2011*). Various theories have been proposed to explain that transition (*Soto and Sonnenschein, 2011*; *Tomasetti and Vogelstein, 2015*; *Jassim et al., 2023*; *Waclaw et al., 2015*), but none has been tested in vivo at the single-cell level. Cancer initiation is thus believed to be a *rare* event taking place at the level of *individual cells* (*Visvader, 2011*; *Nowell, 1976*; *Hanahan, 2022*; *Pénisson et al., 2022*; *Frumkin et al., 2008*) arising as a result of the accumulation of genetic mutations in so-called Mut-driver genes (oncogenes such as KRAS [*Malumbres and Barbacid, 2003*; *Pylayeva-Gupta et al., 2011*], tumor suppressor genes [*Strachan and Read, 1999*] such as TP53 or lifespan [*Blasco, 2005*] genes such as TERT). Notably, mutant KRAS is the most frequent driver of several cancers (*COSMIC,*

*For correspondence:
pierluigi.scerbo@phys.ens.fr (PS);
david@lps.ens.fr (DB);
bertrand.ducos@ens.fr (BD)

Competing interest: The authors declare that no competing interests exist.

*2024*; *Punekar et al., 2022*): about 27% of all human cancers, 45% of colorectal, and 90% of pancreatic cancers (*Merz et al., 2021*).

Recently, it has been shown that genes involved in embryonic development, pluri/multipotency, and cell reprogramming such as VENTX/NANOG and POU5/OCT4 are abnormally reactivated in late cancer stages, where acting as Epigenetic Drivers (Epi-Drivers) they empower cancer cells with cancer stem cell (CSCs) features, resistance to anticancer therapies, and potential for cancer recurrence/relapse (*Vogelstein et al., 2013*; *Hepburn et al., 2019*; *Villodre et al., 2016*; *Ducos et al., 2022*; *Rawat et al., 2010*; *Laise, 2022*; *Park et al., 2021*). Although it is evident that such Epi-Drivers confer a selective advantage to CSCs in a full-blown cancer, whether they play a role during the early phases of malignant transformation is still unknown.

Current approaches to the study of cancer use constitutive or conditional expression of Mut-driver (or Epi-driver genes; *Shibata et al., 2018*) in specific tissues, i.e., in many cells, even though only a small subset of these cells eventually leads to the growth of tumors (*Adashek et al., 2020*), often observed when the tumor already consists of many thousands of heterogeneous abnormal cells. Hence, carcinogenetic processes observed among sibling organisms (*Baggiolini et al., 2021*; *Ablain et al., 2018*; *Kaufman et al., 2016*) occur with variable latency period from the onset of induction, at different locations and develop asynchronously. As a result, the initial stages of tumorigenesis are difficult to study, the state of the cell(s) of origin difficult to assess and control while its cellular environment is perturbed by the induced expression of the Mut- or Epi-driver genes. Due to the rarity of the event in vivo (*Kaufman et al., 2016*), a statistically relevant single-cell tracking and characterization of the early stages of tumorigenesis has therefore never been done. This emphasizes the need to predict or control the cell undergoing malignant transformation in vivo in order to pave the way for a study of the cellular and molecular events involved in the initial stages of tumorigenesis.

To address these issues, we have developed an optogenetic approach to control the expression of an oncogene in a single cell (*Feng et al., 2017*), with the goals of: (1) measuring the probability of the malignant transformation of a single cell in a live organism in various backgrounds and (2) tracking and characterizing the development of a tumor from the original cell. In the following we show the synergy between only two factors: the oncogene kRasG12V and a reprogramming factor (Ventx, Nanog, or Oct4) increases the probability of carcinogenesis from a single cell by many orders of magnitude when compared to the expression of either of these genes (or none).

Optogenetics approaches allow for the photocontrol and monitoring of the activity of biomolecules in vivo. The approach we developed uses a photoactivable analog of tamoxifen (caged cyclofen [cCYC]) to control the activity of proteins fused to the ERT-receptor (*Sinha et al., 2010*; *Zhang et al., 2018*) (a modified estrogen binding domain; *Feil et al., 1996*; *Figure 1A*). These protein constructs are sequestered by cytoplasmic chaperones. Once cCYC is uncaged by light (with one-photon illumination at ~375 nm or two-photon at ~750 nm), cyclofen (CYC) is released (*Sinha et al., 2010*). It binds to the ERT-receptor and releases the fused protein (e.g. a Cre-ERT recombinase) from its complex with cytoplasmic chaperones (*Zhang et al., 2018*; *Figure 1A*).

## Results

We used this approach to photocontrol the activity of a Cre/loxP recombination system in a transgenic zebrafish line (Tg*actin*:loxP-EOS-stop-loxP-KRASG12V-T2A-H2B-mTFP; *ubi*:Cre-ERT; *myl7*:EGFP) carrying a floxable (loxP flanked) EOS gene (coding for a green fluorescent protein) upstream from the oncogene (*Feng et al., 2017*; *Sinha et al., 2010*) (KRASG12V). In such a transgenic zebrafish line hereafter referred to as KRASG12V line (*Figure 1B*, *Figure 1—figure supplement 1B*), the expression of EOS can be switched to KRASG12V and H2B-mTFP (nuclear blue fluorescence) upon 1 hr incubation in cCYC, followed by washing and UV (365 nm) cCYC uncaging which triggers CRE-ERT activation (*Figure 1C*); mTFP fluorescence can be observed about 30 min post-illumination (*Figure 1—figure supplement 1B*) and is stably maintained in zebrafish (*Figure 1G*, *Figure 1—figure supplement 1C*). Consistent with an acquired refractory cell state and a loss of oncogenic competence (*Baggiolini et al., 2021*), whole body expression of the oncogene at 1 day post-fertilization (dpf) did not result in tumorigenesis. Notice that consistent with previous studies (*Feng et al., 2017*; *Sinha et al., 2010*; *Zhang et al., 2018*), where cCYC stability has been validated both in vitro (by chemical analyses) and in vivo (with fluorescent reporters of uncaging), in absence of illumination no spontaneous uncaging of cCYC and thus activation of the oncogene (i.e. blue fluorescent cells) is observed. Notice also that

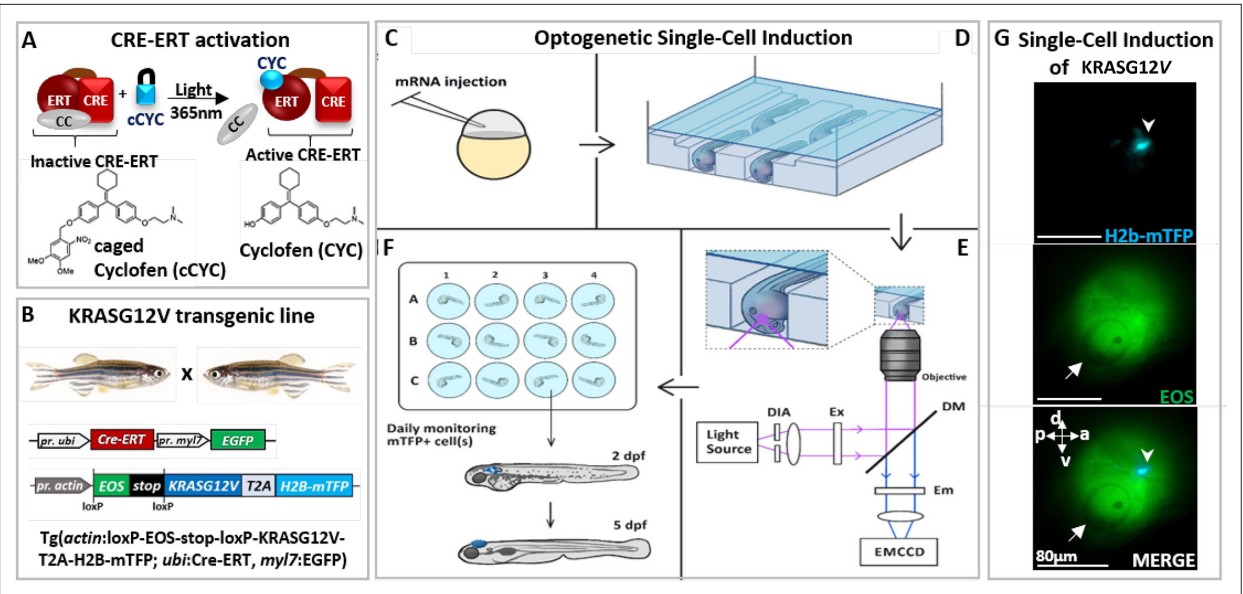

**Figure 1.** Optogenetic setup for single-cell induction. (**A**) Photocontrol of a protein fused to an estrogen receptor (ERT) is achieved by releasing the protein from its complex with cytoplasmic chaperones (CC), upon uncaging of caged cyclofen (cCYC). (**B**) The transgenic zebrafish line engineered to express an oncogene (KRASG12V) upon photoactivation of a CRE-recombinase fused to ERT (CRE-ERT) (as shown in A). (**C**) The mRNAs of Ventx-GR and mRFP (used as a marker) are injected at the one-cell stage. (**D**) At 1 day post-fertilization (dpf) the embryos are mounted in channels in an agarose gel and incubated for 45 min in cCYC on a microscope stage. (**E**) They are washed and illuminated at 405 nm on a microscope to uncage cCYC close to the otic vesicle. A diaphragm (DIA) defines an illumination zone of ~80 μm diameter (see G). Excitation (EX), dichroic mirrors (DM), and emission (EM) filters allow for visualization of Eos and mTFP. The cell in which KRASG12V has been induced is observed within ~1 hr by the fluorescence of mTFP (see G). (**F**) The embryos are transferred into individual wells, incubated overnight in dexamethasone (DEX), washed at 1 day post-induction (dpi) and monitored over the next 5 days. (**G**) Lateral view of zebrafish at ~1 hr post-activation displays a single induced cell (top: blue spot shown by arrowhead in mTFP channel) in the illumination region (middle: Eos channel) in the vicinity of the otic vesicle (white arrow) and bottom: merger of both channels. Body axes (a: anterior; p: posterior; d: dorsal; v: ventral) are shown.

The online version of this article includes the following source data and figure supplement(s) for figure 1:

**Figure supplement 1.** Characterization of the double transgenic line Tg(β-*actin*:loxP-EOS-stop-loxP-KRASG12V-T2A-H2B-mTFP; *ubi*:Cre-ERT; *myl7*:EGFP).

**Figure supplement 2.** Transient activation of Ventx reprogramming factor.

**Figure supplement 3.** Synergy between a reprogramming factor and KRASG12V oncogene efficiently induces tumors in zebrafish larvae.

**Figure supplement 4.** Synergy between the Ventx reprogramming factor and the KRASG12V oncogene induces tumors in zebrafish larvae.

**Figure supplement 5.** KRASG12V *plus* Ventx induces a cancer-like gene expression signature.

**Figure supplement 5—source data 1.** Reverse transcriptase quantitative PCR (RT-qPCR) raw data.

**Figure supplement 5—source data 2.** Data file of the heatmap shown in *Figure 1—figure supplement 5*.

**Figure supplement 6.** KRASG12V *plus* Ventx induces a cancer-like gene expression signature.

once cCYC is uncaged embryos are maintained in E3 medium with no CYC and thus no activation of the oncogene.

To test for the possible impact of a reprogramming factor on the malignant transition, embryos from this transgenic line were injected at the one-cell stage with the mRNA of a construct consisting of a glucocorticoid receptor (GR) fused with a reprogramming factor (Ventx, Nanog, or Oct4; *Ducos et al., 2022*; *Scerbo and Monsoro-Burq, 2020*; *Figure 1—figure supplement 2A*). The resulting protein (e.g. Ventx-GR) is sequestered by cytoplasmic chaperones and transiently activated upon incubation of zebrafish in dexamethasone (DEX) (*Figure 1—figure supplement 2B*). Activation of the oncogene (*Feng et al., 2017*) at 1 dpf, if and only if followed by transient activation of Ventx (or Nanog or Oct4), did yield reproducible hyperplasic outgrowths in many different tissues (including brain, intestine, pancreas, and liver; *Figure 1—figure supplement 3A–C*). Immunohistochemistry detection of phospho-ERK activity (*Figure 1—figure supplement 4A and B*), hematoxylin and eosin (H&E) staining (*Figure 1—figure supplement 4C*), decreased survival (*Figure 1—figure supplement 4D*),

and reverse transcriptase quantitative PCR (RT-qPCR) of selected genes (*Figure 1—figure supplement 5*; *Figure 1—figure supplement 6A and B*) all display features associated with tumorigenesis. None of the controls (activation of kRasG12V only, Ventx-GR only, incubation in CYC or DEX, etc.) developed hyperplasia, but rather grew into normal zebrafish (*Figure 1—figure supplement 1B and C*; *Figure 1—figure supplement 3A*). These controls imply that the observed tumor is only a result of the joint activation of kRasG12V (permanently) and a reprogramming factor (transiently).

Since both kRasG12V and Ventx have been reported to play a role in brain cancer (*Lam et al., 2022*), we decided to activate the oncogene kRasG12V at 1 dpf in a single normal cell of the brain of a transgenic zebrafish (injected with the mRNA of Ventx-GR at one-cell stage). Activation of the oncogene in a single cell was achieved by 1 hr incubation in cCYC, followed by washing and illumination at 405 nm to uncage cCYC in a small area (diameter ~80 μm) of the brain (in the vicinity of the otic vesicle) (*Figure 1G*). Notice that following uncaging the embryos are maintained in a CYC-free E3 medium.

Subsequent to cCYC uncaging, with probability ~50% the oncogene is activated in a single cell of the illuminated area (activation statistics is shown in *Figure 2A*), identified within ~1 hr by the blue fluorescence of its nuclear marker (H2B-mTFP), see *Figures 1G and 2A*, *Figure 2—figure supplement 1*. H2B-mTFP positive cells are only observed at the site of illumination (i.e. brain), attesting to the precision of our optogenetic method (*Figure 2A*). This observation specifies whether one, two, or more cells were activated (*Figure 2—figure supplement 1*) within ~1 hr post-illumination. Notice that activation of the oncogene alone does not give rise to cancer (the activated cell usually disappears within a few days, *Figure 2—figure supplement 2A*), which implies that the proliferation of kRasG12V positive (mTFP$^+$, blue fluorescent) cells - reported next - is not due to late activation of the oncogene (and its fluorescent marker).

Indeed, if and only if following the local expression of the oncogene, Ventx-GR is transiently activated (by 24 hr incubation in DEX) does the cell divide and proliferate (*Figure 2B and C*, *Figure 2—figure supplement 3*, *Figure 2—figure supplement 4*). We observed that at 1 day post-induction (dpi) ~50% of the induced cells had divided and expanded clonally by a factor ~3 (*Figure 2B*). In the other 50% of cases, the activated single cell had neither divided nor died (*Figure 2B*). Surprisingly, at 3–5 dpi, we observed in all zebrafish larvae (n=30) that the induced cell(s) gave rise to progeny that display short-range dispersion (*Figure 2—figure supplement 4A and B*) and then give rise to a tumor mass in the brain with local infiltration of malignant cells (*Figure 2C*). H&E staining of the brain tissue (*Figure 2D and E*) and metastases (*Figure 3*) further confirm the malignant state of the induced cell and its progeny.

In parallel with tumor mass formation in the brain (*Figure 2*), we observed that some cells of the progeny re-localized to new loci far from the brain, such as the heart (*Figure 3A*), the digestive tract (*Figure 3B*; *Figure 2—figure supplement 4D*), and the trunk (*Figure 2—figure supplement 3B*; *Figure 2—figure supplement 4C*). We therefore deduce that the transient activation of Ventx alters the state of a cell expressing a mutated oncogene (i.e. KRASG12V) and induces its malignant transformation in vivo, with tumorigenic potential and the capacity to generate invasive progeny. Notice that metastatic cells must be the progeny of the initial (blue fluorescent) cell as the larvae are incubated in embryo medium with no cCYC. Furthermore, absent VentX activation we do not observe proliferation and metastatic behavior, rather the initially induced cell disappears, see *Figure 2—figure supplement 2A*.

Of the larvae in which a few (1–6) cells were expressing the oncogene and in which Ventx was transiently activated (KR+VX cell), all (N=30) developed tumors within 5 dpi. The frequency of tumor development is therefore $F_1$=1. Conversely, the probability for such a cell to not develop a tumor is $F_0$=0. Due to the finite size of the sample, we estimate the probability of tumor development from a single cell to be: $P_1$>80% ($\chi^2$=3.75; df = 1).

To definitely confirm the carcinogenic nature of the KR+VX-induced cells, we isolated and injected a single cell from a hyperplasic tissue (identified by the blue fluorescence of its nucleus) into a naïve host zebrafish larva. This led to integration, migration, and colonization of the host tissues by the progeny of the transplanted cell (*Figure 4A and B*), with strong pERK activity detected in tumor masses of the host (*Figure 4B*). Out of 52 transplanted host zebrafish, 31 developed tumors, a probability of tumor development (60%) consistent with previously reported efficiency of tumor cells transplantation in zebrafish (*Campbell et al., 2021*). The capacity of the progeny of the induced cells to metastasize and

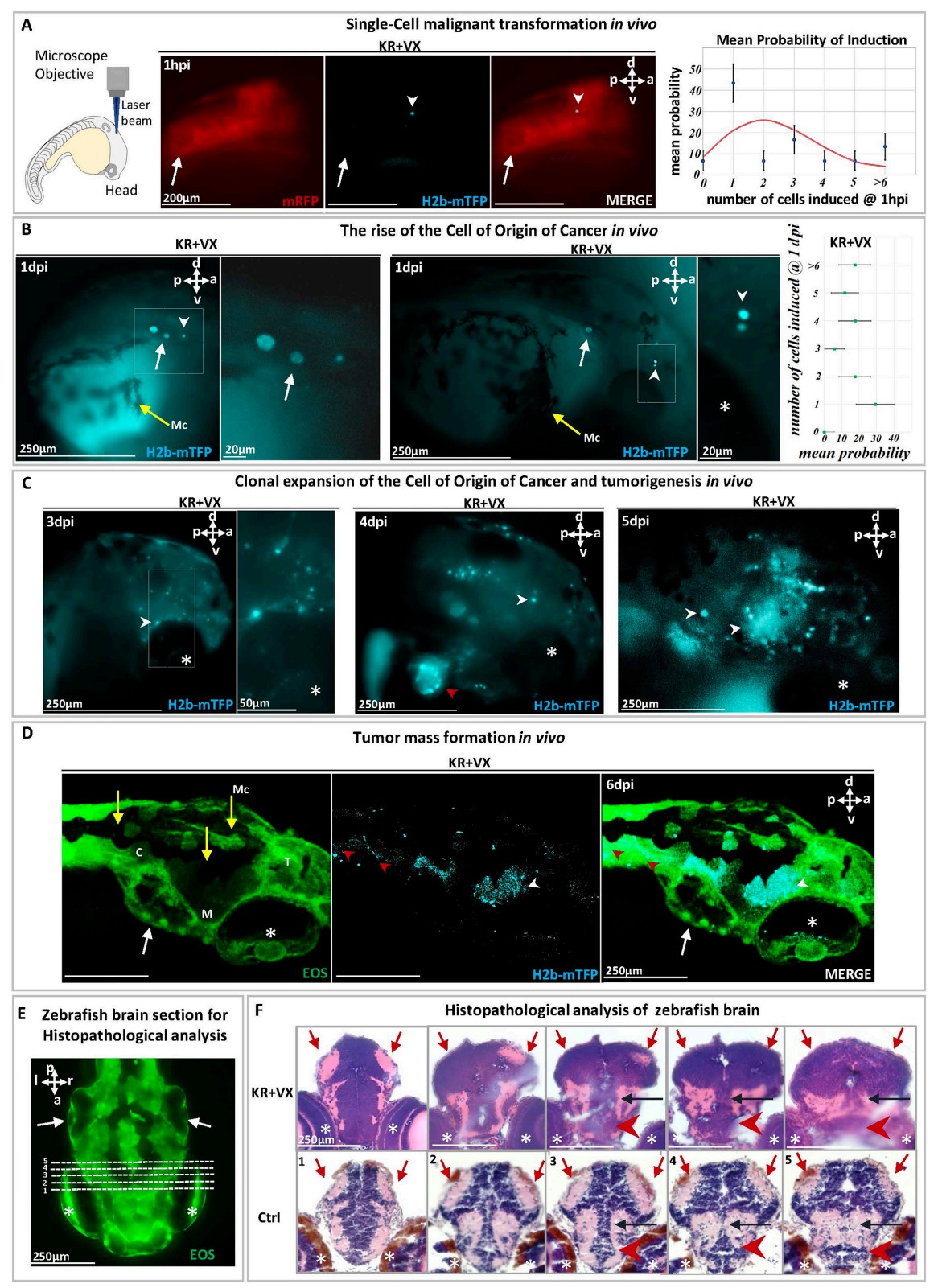

**Figure 2.** Malignant transformation of a single cell triggering carcinogenesis in vivo. (**A**) At 1 day post-fertilization (dpf), a single cell in a zebrafish brain was photoinduced to express the oncogene KRASG12V, identified (white arrowhead) within ~1 hr by the blue fluorescent H2B-mTFP. Membrane-bound mRFP is used as tracer. Transient (24 hr) dexamethasone (DEX) activation of Ventx is done following photoactivation. The probability p of inducing one or more (blue fluorescent) cells is shown on the right panel, together with the Poisson distribution (red curve) expected for the independent induction of

*Figure 2 continued on next page*

*Figure 2 continued*

k cells (error bars are statistical errors on the mean: $\sigma = \sqrt{p\left(1-p\right)/N}$, where N is the total number of observed embryos). (**B**) At 1 day post-induction (1 dpi), the activated cell may have divided (middle panel) giving rise to two mTFP$^+$ (blue fluorescent) cells (white arrowhead) or may not have divided (left panel). The probability of observing k blue fluorescent cells at 1 dpi is shown on the right panel. (**C**) At 3 dpi the original cell expanded clonally (white arrowheads) by short-range dispersal within the brain. At 4 dpi the brain has been colonized (middle panel) by the progeny of the activated cell that display tumor growth as well as dispersal in the head or entering into the cardiovascular system (red arrowhead). At 5 dpi (right panel), a tumor mass is formed. (**D**) Confocal microscopy of a larval head displaying tumors (white arrowheads) and dispersal in the trunk (red arrowheads). (**E**) Histopathological sections of larval brain (dorsal view). Dashed lines (1–5) indicate sections shown in (**F**). Hematoxylin and eosin (H&E) staining of brain at 5 dpi of KR+VX-induced larva (depigmented) is compared to normal brain (Ctrl, melanocytes in brown). At 5 dpi, the optic tectum (red arrows), the tegmentum (black arrow), and hypothalamus (red arrowhead) are infiltrated by a dysplastic tumor, progeny of the initial induced single cell. An asterisk (*) indicates the eye and a white arrow the otic vesicle. Scale bars and the body axes (a: anterior; p: posterior; d: dorsal; v: ventral) are shown. T=telencephalon; M=mesencephalon; C=cerebellum; Mc =melanocytes (yellow arrows).

The online version of this article includes the following figure supplement(s) for figure 2:

**Figure supplement 1.** Photoactivation of one or more cells by caged cyclofen (cCYC) uncaging.

**Figure supplement 2.** Single-cell activation of the KRASG12V oncogene is not sufficient to initiate carcinogenesis.

**Figure supplement 3.** Tracking of the clonal expansion of a photoinduced cell in one embryo over 7 days.

**Figure supplement 4.** Metastatic spreading following activation of KRASG12V and Ventx (KR+VX).

develop a tumor mass when transplanted in a naïve zebrafish is - to the best of our knowledge - the operational definition of a malignant tumor. This result definitely confirms the cancerous nature of cells expressing a mutated oncogene and exposed to the transient activation of Ventx, with subsequent alteration of the homeostasis. Notice that injection into a naïve larva of an oncogene expressing cell (identified by its nuclear blue fluorescence) which did not experience the transient activation of a reprogramming factor does not yield a tumor and disappears within a few days (*Figure 2—figure supplement 2B*).

## Discussion

The surprising frequency of somatic mutations occurring in physiologically normal tissues (*Martincorena and Campbell, 2015*; *Brash, 2015*) raises the question of what combinations of events are sufficient for the malignant transformation of a single cell and the rise of the cell of origin of cancer. The robust deterministic process of single-cell cancer induction that we uncovered suggests that the aberrant reactivation of Epi-Driver genes involved in reprogramming/pluripotency (such as VENTX/ NANOG, POU5/OCT4) might be relevant to the irreversible malignant transformation of a cell. Reprogramming Epi-Drivers are important regulators of cell viability, survival, and proliferation in several cellular contexts, from embryonic stem cells to cancer cells (*Hepburn et al., 2019*; *Villodre et al., 2016*; *Ducos et al., 2022*; *Rawat et al., 2010*; *Laise, 2022*; *Park et al., 2021*). Due to their capacity to modulate epigenetic memory and cell plasticity, these reprogramming factors may drive the early stages of malignant transformation in vivo once (re)activated aberrantly. They likely share some mechanism(s) with processes such as induced nuclear reprogramming (*Hepburn et al., 2019*; *Villodre et al., 2016*; *Ducos et al., 2022*), pluripotency maintenance, and/or endogenous cell reprogramming during development (*Scerbo and Monsoro-Burq, 2020*). Thus, consistent with our results, the reactivation (*Scerbo and Monsoro-Burq, 2020*) of the Neural Crest (NC) Progenitor program (possibly via the stochastic expression of VENTX/NANOG and/or POU5/OCT4) has been shown in BRAF/p53 double mutant cells to yield NC-related tumors in vivo (*Baggiolini et al., 2021*; *Ablain et al., 2018*; *Kaufman et al., 2016*). We surmise that activation of Epi-Driver genes in NC-related tumors might confer predictable tumorigenesis.

Since cells carrying cancer-causing mutations do not deterministically developed cancers in vivo (*Kaufman et al., 2016*; *Campbell et al., 2021*), we suggest that the probability of a mutated cell to enter into malignant transformation, or to maintain its physiological functions despite mutations, correlates with the probability of the reactivation of reprogramming factors. Our results support a 'Vogelgram' (*Fearon and Vogelstein, 1990*) two-step model (*Figure 5*) for irreversible single-cell malignant transformation in vivo, mirroring the two hits hypothesis proposed by *Berenblum and Shubik, 1949*.

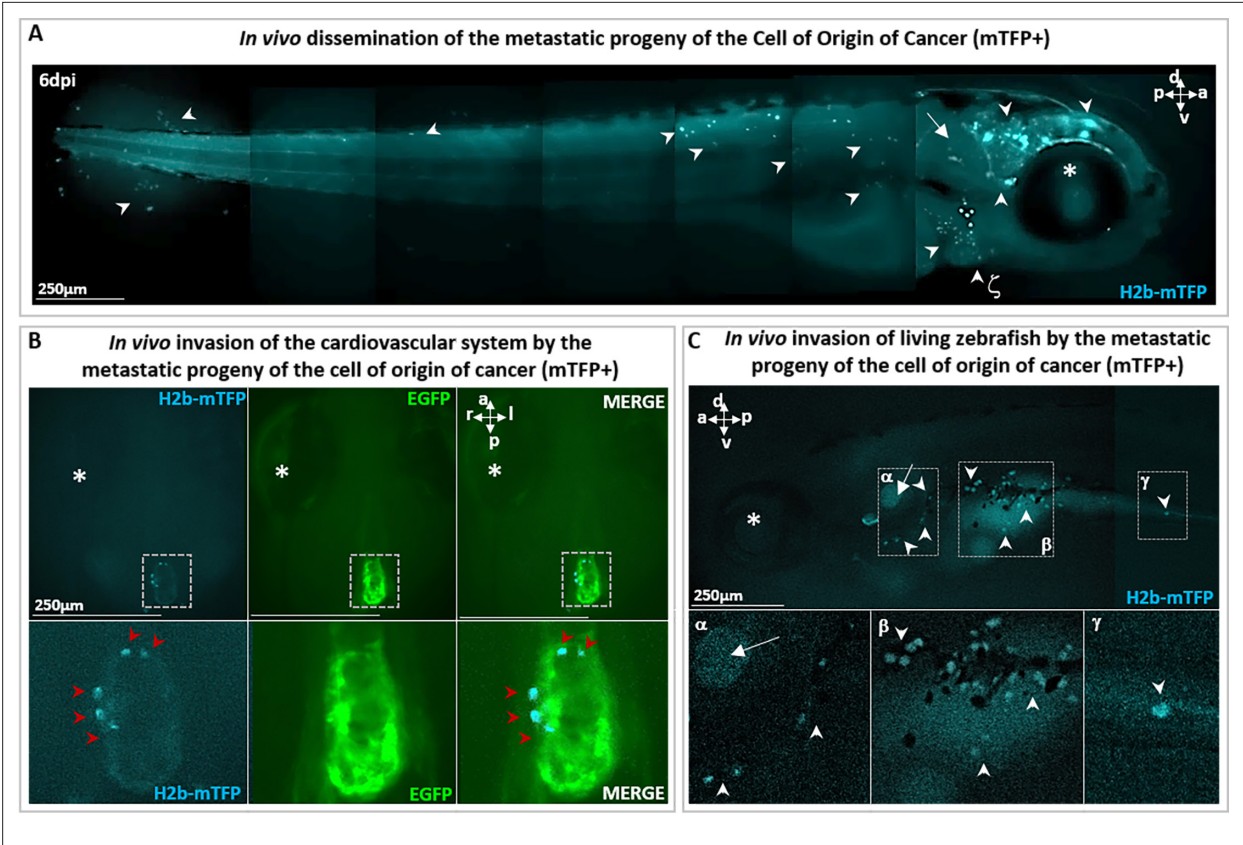

**Figure 3.** Metastatic cells following single-cell malignant transformation. (**A**) A zebrafish larva, in which a single cell in the brain was photoinduced (at 1 day post-fertilization [dpf]) to express the oncogene KRASG12V by the blue fluorescence of the expression marker (H2b-mTFP) as in *Figure 2*, shows that the progeny of the cell of origin of cancer give rise to a tumor mass in the brain (white arrowhead), as well as to migrating metastatic-like cells that disseminate in the whole organism (white arrowhead), some localizing in proximity of arterial branchial arches (indicated by ζ and white arrowhead), trunk, and tail fin. (**B**) H2b-mTFP positive cells can migrate far from the site of induction (brain) and colonize ectopic tissues located in the heart and (**C**) the bottom of otic vesicle (in proximity of the primary head sinus, designated by α), the digestive tract (designated by β and γ), a feature characteristic of metastatic cancer cells. Note that no anesthetics (e.g. tricaine) or mounting media (low melting point agarose or methylcellulose) were used to block live zebrafish, during both monitoring and imaging performed on live zebrafish. An asterisk (*) indicates the eye and a white arrow the otic vesicle. Scale bars and body axes (a: anterior; p: posterior; d: dorsal; v: ventral) are indicated.

Thus, the appearance of Mut-Drivers in a normal (preprocancer; *Brash, 2015*) cell (*Martincorena and Campbell, 2015*) within healthy tissue would act as a permissive but insufficient initiator for malignant cell transformation. Despite the frequency of mutational insults, which are often caused by external cues (chemical carcinogens, UV, etc.) or senescence/aging, it is known that surveillance mechanisms like cell competition between wild-type and mutated cells safeguard tissue homeostasis and results in active elimination of mutant cells from the tissue (*Brown et al., 2017*; *Porazinski et al., 2016*; *van Neerven and Vermeulen, 2023*). Our data imply that the aberrant reactivation of Epi-Driver genes involved in reprogramming/pluripotency may promote the irreversible and deterministic malignant transformation in the cell of origin of cancer leading to carcinogenesis.

Our hypothesis is further strengthened by the observation that the variation in cancer risk among tissues can be related to the number of stem cell divisions and tissue renewal (*Tomasetti and Vogelstein, 2015*). The aberrant reactivation (or maintenance) of Epi-Driver genes in these regenerative events, in cooperation with incident mutations, might be the key deterministic factor in the switching of a normal/healthy cell into the first cell of origin of cancer.

Furthermore, the results reported here are compatible with the recently proposed 'ground state theory of cancer initiation' (*Jassim et al., 2023*). According to that theory a malignant transformation may occur in a cell harboring an oncogenic mutation upon a change of its functional state (its 'ground state'). Here, the transient activation of a reprogramming factor (i.e. Ventx, Nanog, or Oct4) possibly

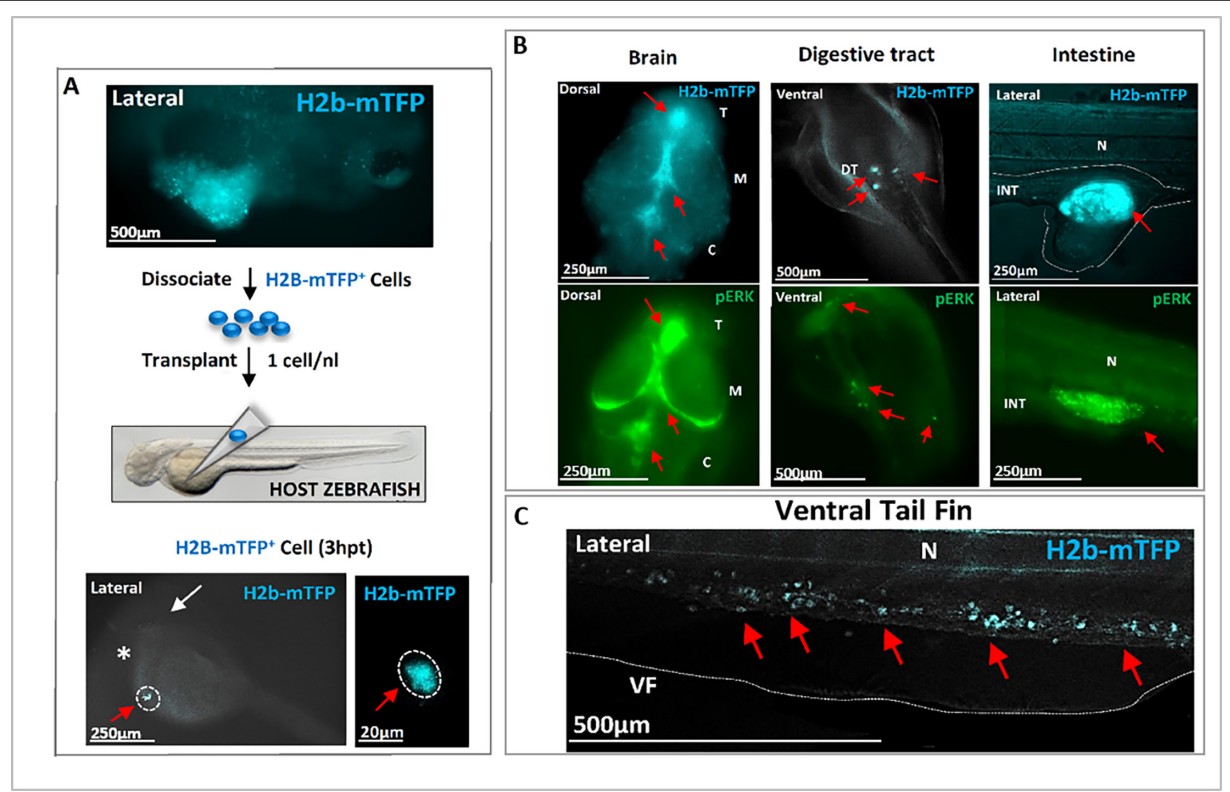

**Figure 4.** Transplantation of single cell(s) from hyperplasic tissue reveals cancer-initiating potential. (**A**) Following KRASG12V *plus* Ventx activation, hyperplasic tissues (blue fluorescent) are observed at 6 days post-fertilization (dpf), top figure. The cells of the hyperplasia were dissociated and isolated blue (KRAS expressing) cells were transplanted (≈1 cell *per* host) at 2 dpf in a *Nacre* (*mitf* -/-) zebrafish line for a better tracking of the transplanted cells. The transplanted H2b-mTFP positive (blue) cell (red arrow) can be visualized as early as 3 hr post-transplantation (3 hpt) in the yolk of the host, close to the duct of Cuvier. White arrow indicates the head/eye (lateral view). (**B**) At 3 dpt the blue cell(s) from the hyperplasic tissue of the donor have colonized the host *Nacre* zebrafish larvae. Tumors in the brain, digestive tract, and intestine are observed and characterized by the blue fluorescence of the donor KRAS expressing cells (red arrows; n=31 out of 52 host individuals). In the bottom, immunofluorescence (IF) analysis of representative host zebrafish larvae with specific high level of phosphorylated ERK activity (pERK, red arrows) in the brain, intestine, and digestive tract. (**C**) A high number of exogenous blue fluorescent cells are here observed to migrate in the tail (red arrows). These observations indicate that the transplanted founder cell has both migratory, colonizing behavior, as well as survival growth advantage in the host to form tumors, and thus to re-initiate carcinogenesis. Scale bars and body axes (a: anterior; p: posterior; d: dorsal; v: ventral) are shown; T=telencephalon; M=mesencephalon; C=cerebellum; N=notochord; VF = ventral fin.

synergizes with the oncogene to alter the epigenetic and functional state of the cell allowing its transformation into a tumorigenic cell.

This may also explain the apparent discrepancy between the present deterministic and rapid tumor development upon activation in a single cell of both kRasG12V and a reprogramming factor and the reports of stochastic and random tumor development over weeks following activation of kRasG12V in a whole tissue (brain [*Ju et al., 2015*], pancreas [*Oh and Park, 2019*], or intestine [*Lu et al., 2018*]). In the latter case, transformed cells may have been triggered to undergo a malignant transformation by the stochastic activation of a reprogramming factor/Epi-Driver, which is consistent with the recent finding that cancer initiation might occur by transient perturbations of epigenetic regulators (*Parreno et al., 2024*).

Importantly, our observation that the progeny of the cell(s) of origin of cancer display a short-range dispersal within the tissue during the earliest phases of tumorigenesis and prior to the effective appearance of tumor mass corroborates a recent theoretical model predicting such an early dispersal during carcinogenesis to explain heterogeneity, therapeutic resistance, and tumor relapse (*Waclaw et al., 2015*).

Being the first experiments to predictably and noninvasively control the malignant transformation of a single cell in an unaltered microenvironment, our approach opens a new vista on the study of 'the

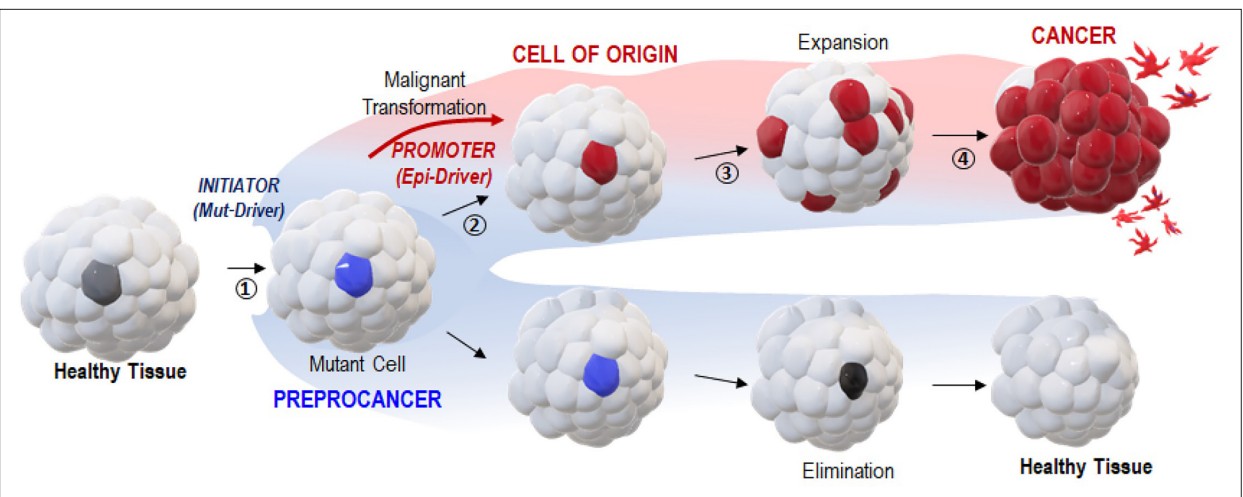

**Figure 5.** A two-step 'Vogelgram' model of deterministic and irreversible single-cell malignant transformation in vivo. Within a healthy tissue, a normal cell (in light gray) acquires (step 1) genetic mutation(s) in driver oncogene(s) (Mut-Drivers such as kRASG12V). Such a mutant cell (in blue, referred to as a preprocancer [**Brash, 2015**] cell) can (step 2) aberrantly activate the expression of Epi-Drivers involved in pluripotency/reprogramming (e.g. VENTX/ NANOG, POU5/OCT4) thus undergoing deterministic and irreversible malignant cell transformation (red cell, the cell of origin of cancer); (step 3) in situ short-range dispersal of the early malignant cells and (step 4) further progression to cancer mass and to the appearance of metastatic cells. Inversely, the mutant (preprocancer) cell (in blue) can maintain its physiological functions and be eventually eliminated from the healthy tissue.

cell of origin of cancer' (**Blanpain, 2013**), an acknowledged enigma of cancer research. As such our results raise many questions: what mechanisms are at work in the transition from the preprocancer state to a tumorigenic state? Is the aberrant reactivation of reprogramming factors a key step in the initiation of carcinogenesis, independent of age, tissue, or oncogenic mutation? Do drugs against reprogramming factors reduce the probability of carcinogenesis? Conversely, do some carcinogens (e.g. environmental pollutants, pesticides, endocrine disruptors, etc.) act by reactivating those factors?

By allowing for specific and reproducible single-cell malignant transformation in vivo, our optogenetic approach opens the way for a quantitative study of the initial stages of cancer at the single-cell level (e.g. tracking and characterization), that will allow one to address many of these questions.

## Materials and methods
### Fish lines and maintenance

Zebrafish (*Danio rerio*) were raised and maintained in an approved Fish Facility (C75-05-32 at IBENS) on a 14–10 hr light-dark diurnal cycle with standard culture methods (**Westerfield, 2000**). Embryos collected from natural crosses were staged according to **Kimmel et al., 1995**. The Tg(*actin*:loxP-EOS-stop-loxP-KRASG12V-T2A-H2B-mTFP) was generated by injecting the plasmid pT24-*actin*:loxP-EOS-stop-loxP-KRASG12V-T2A-H2B-mTFP, which contains the homologous cDNA sequence of KRASG12V from human, with tol2 mRNA transposase. Founder transgenic fish were identified by global expression of Eos. The Tg(*ubi*:Cre-ERT; *myl7*:EGFP) was previously described (**Mosimann et al., 2011**). The double transgenic line Tg(*actin*:loxP-EOS-stop-loxP-KRASG12V-T2A-H2B-mTFP; *ubi*:Cre-ERT; *myl7*:EGFP) was created by crossing Tg(*actin*:loxP-EOS-stop-loxP-KRASG12V-T2A-H2B-mTFP) and Tg(*ubi*:Cre-ERT; *myl7*:EGFP). Founder double transgenic fish were selected by global expression of Eos and expression of EGFP in the developing heart. Zebrafish were imaged for phenotypic analysis throughout early development, from embryonic to larval stage, and then fixed (at 6 dpf) with PAXgene Tissue Container Product (QIAGEN) for RT-qPCR or with 4% PFA (Thermo Fisher) for histological (H&E) analyses.

### cCYC treatment and UV uncaging

To induce kRASG12V expression, 24 hpf embryos were incubated in 6 µM cCYC for 1 hr (in the dark), rinsed in E3 medium followed by 5 min photoactivation with an ~365 nm UV lamp (measured intensity: 4 mW/cm²). They were then transferred back to E3 medium (with no cCYC)+10 µM DEX (to release

Ventx-GR from its complex with cytoplasmic chaperones). The embryos at 1 dpi were washed 3× in E3 medium to remove DEX. From thereon embryos are exposed neither to cCYC nor to DEX. For this experiment, we used a benchtop UV lamp (Fisher VL-6-L) which emits a peak wavelength at 365 nm with a full width at half maximum (FWHM) of 40 nm and delivers on the illuminated sample a typical photon flux of $\sim 1.25 \times 10^{-4}$ Einstein/(s·m$^2$).

Localized uncaging was performed by illumination for 7 min on a Nikon Ti microscope equipped with a light source peaking at 405 nm, *Figure 1*. The size of the uncaging region was controlled by an iris that defines a circular illumination of diameter ~80 µm. After 0.5–1 hr following uncaging the illuminated region of each embryo was imaged to identify and count mTFP positive cell(s) (cells in which the oncogene and its reporter fluorophore was activated). In about 50% of cases one cell was thus photoactivated, the other ~50% corresponded to more than one activated cell, see *Figure 2A*. Embryos were transferred to a 12-well plate with E3+10 µM DEX, incubated in total darkness over-night and washed 3× with E3 at 1 dpi (to remove DEX).

## DEX induction

Transgenic zebrafish were injected at the one-cell stage with the mRNA of a gene-construct Ventx-GR (*Xenopus* ventx2, 450 pg), Nanog-GR (mouse Nanog, 100 pg), or Pou5/Oct4 (*Xenopus* pou5f3.1/oct91, 100 pg) together with a red fluorescent expression marker mRFP (50 pg). The protein products of these constructs are sequestered by cytoplasmic chaperones and released upon incubation of the embryos in a medium containing 10 µM DEX. In single-cell activation experiments, zebrafish were selected from their strong intensity of fluorescent mRFP signal for local activation of kRasG12V via uncaging of 6 µM cCYC (a gift of I Aujard and L Jullien).

## Reverse transcriptase quantitative PCR

Zebrafish larvae were fixed (as mentioned above) at 6 dpf. Total RNAs were extracted using the RNeasy micro kit (QIAGEN) according to the manufacturer's protocols. Sample quantity and purity, reverse transcription, pre-amplification, and high-throughput qPCR were performed as in *Zhang et al., 2022*.

## Reverse transcription

cDNA synthesis was performed using Reverse Transcription Master Mix from Standard Biotools according to the manufacturer's protocol with random primers in a final volume of 5 µL containing 40 ng total RNA using a Nexus thermocycler (Eppendorf). cDNA samples were diluted by adding 20 µL of low TE buffer (10 mM Tris; 0.1 mM EDTA; pH = 8.0 [TEKNOVA]) and stored at –20°C.

## Specific target amplification

1.25 µL of each diluted cDNA was used for multiplex pre-amplification with Standard Biotools PreAmp Master Mix at 18 cycles. In a total volume of 5 µL, the reaction contained 1 µL of pre-amplification mastermix, 1.25 µL of cDNA, 1.25 µL of pooled TaqMan Gene Expression assays (Life Technologies, Thermo Fisher) with a final concentration of each assay of 180 nM (0.2×) and 1 µL of PCR water. The cDNA samples were subjected to pre-amplification following the temperature protocol: 95°C for 2 min, followed by 18 cycles at 95°C for 15 s and 60°C for 4 min. The pre-amplified cDNA were diluted 5× by adding 20 µL of low TE buffer (TEKNOVA) and stored at –20°C before qPCR.

## High-throughput real-time PCR

Quantitative PCR was performed using the high-throughput platform BioMark HD System and the 48.48 GE Dynamic Arrays (Standard Biotools). 6 µL of sample master mix (SMM) consisted of 1.8 µL of 5× diluted pre-amplified cDNA, 0.3 µL of 20× GE Sample Loading Reagent (Standard Biotools), and 3 µL of TaqMan Gene Expression PCR Master Mix (Life Technologies, Thermo Fisher). Each 6 µL assay master mix (AMM) consisted of 3 µL of TaqMan Gene Expression assay 20× (Life Technologies) and 3 µL of 2× Assay Loading Reagent (Standard Biotools). 5 µL of each SMM and each AMM premixes were added to the dedicated wells. The samples and assays were mixed inside the chip using MX-FC controller (Standard Biotools). Thermal conditions for qPCR were: 25°C for 30 min and 70°C for 60 min for thermal mix; 50°C for 2 min and 95°C for 10 min for hot start; 40 cycles at 95°C for 15 s and 60°C for 1 min. Data were processed by automatic threshold for each assay, with linear derivative

baseline correction using BioMark Real-Time PCR Analysis Software 4.0.1 (Standard Biotools). The quality threshold was set at the default setting of 0.65.

RT-qPCR measurements were done in triplicate on pooled zebrafish larvae and single zebrafish larva (for KR+VX condition), as indicated in the text (see *Figure 1—figure supplement 5—source data 1*). Normalization and quantification were obtained with the ΔΔCt method using *rpl13a* as a reference gene. The relative expression of the genes R under the different conditions analyzed was calculated as follows using the method described by *Livak and Schmittgen, 2001*:

$$R = 2^{-\Delta\Delta Ct} \text{ where } \Delta\Delta Ct = (Ct_{\text{gene of interest}} - Ct_{\text{reference gene}})_{\text{test condition}} - (Ct_{\text{gene of interest}} - Ct_{\text{reference gene}})_{\text{control condition}}.$$

## Immunohistochemistry

The zebrafish obtained in the various conditions and stages mentioned in the text were fixed in 4% PFA overnight at 4°C, followed by dehydration with 100% methanol at −20°C for more than 1 day. After gradual rehydration of methanol and wash with PBS/Tween 0.1%, the embryos were incubated in a blocking solution: 1% Triton, 1% DMSO, 1% BSA, and 10% sheep serum (Sigma) in PBS on a shaker for 1 hr at room temperature, followed by incubation on a shaker overnight at +4°C with 1:500 anti-phospho-Erk antibody (phospho-p44/42 MAPK (Erk1/2) (Thr202/Tyr204) (D13.14.4E) XP). Rabbit mAb detects endogenous levels of p44 and p42 MAP Kinase (Erk1 and Erk2) when dually phosphorylated at Thr202 and Tyr204 of Erk1 (Thr185 and Tyr187 of Erk2), and singly phosphorylated at Thr202 - this antibody does not cross-react with the corresponding phosphorylated residues of either JNK/SAPK or p38 MAP kinases (Ref 4370S Cell Signaling). After washing with PBS/Tween, embryos were incubated overnight at +4°C with 1/1000 secondary fluorescent conjugated antibody, Donkey anti-Rabbit Alexa Fluor 488 (Thermo Fisher A-21206). To visualize the distribution of Ventx-GR in the fish, embryos were incubated with 1:500 anti-HA antibody, tagging the HA sequence fused to Ventx-GR, as the conjugated antibody (Anti-HA, mouse IgG1, clone 16B12 Alex Fluor 488 conjugated, Thermo Fisher Ref A21287) on a shaker overnight at 4°C. All images were taken on a Nikon Ti microscope equipped with a Hamamatsu Orca camera, except for confocal microscopy (*Figure 2C*, *Figure 1—figure supplement 4B*) carried out on a Leica SP5 and Zeiss LSM800 microscope.

## Cell transplantation

Transplantation experiments followed published protocols (*Nicoli and Presta, 2007*). Specifically, zebrafish larvae were grown until 6 dpi, dissociated in dissociation medium (*Bresciani et al., 2018*) (025% trypsin-EDTA *plus* 1/10 Collagenase 100 mg/mL) 10 min at 30°C, mechanically homogenized, resuspended in DMEM-10% Fetal Bovine Serum (FBS), filtered through 70 µm nylon mesh (Corning Cell Strainer Ref 431751), and resuspended in DMEM-10% FBS at 1 cell/1 nL concentration by using LUNA-II cell counter (Logos Biosystems), and injected (~1 nL) at the level of the yolk sac circulation valley (close to the site of the ducts of Cuvier) of host nacre zebrafish at 2 dpf as shown in *Figure 4*.

## Microscopy

Fluorescent images were taken on a Nikon Ti microscope equipped with a Hamamatsu ORCA V2+ camera and a ×10 Plan Fluor objective. Filter setting of CFP: excitation at 438±24 nm, emission at 483±32 nm; mEosFP and Alexa 488: excitation at 497±16 nm, emission at 535±22 nm.

Image analysis was done using ImageJ software.

## Acknowledgements

We acknowledge financial support from the ITMO Cancer of Aviesan within the framework of the 2021–2030 Cancer Control Strategy, on funds administered by Inserm. The gene expression analysis was carried out on the high-throughput qPCR-HD-Genomic Paris Centre core facility and was supported by grants from the Région Ile de France. We are grateful to I Aujard and L Jullien (ENS, Dept. of Chemistry) for a gift of cCYC. We acknowledge useful comments on a draft of this paper by Z Feng, M Distel, and D Louvard. PS acknowledges useful discussions with C Rogard, M Smilla, LAM Scerbo, and EPA Scerbo. We thank the Graphic Atelier of Timon Ducos for scientific illustrations (https://timonducos.com/).

# Additional information

## Funding

| Funder | Grant reference number | Author |
|---|---|---|
| Aviesan ITMO Cancer | | Bertrand Ducos |

The funders had no role in study design, data collection and interpretation, or the decision to submit the work for publication.

## Author contributions

Pierluigi Scerbo, Conceptualization, Investigation, Methodology, Writing – original draft, Writing – review and editing; Benjamin Tisserand, Methodology; Marine Delagrange, Data curation, Software, Formal analysis, Methodology; Héloise Debare, Data curation, Software, Methodology; David Bensimon, Conceptualization, Formal analysis, Supervision, Methodology, Writing – original draft, Project administration, Writing – review and editing; Bertrand Ducos, Conceptualization, Resources, Data curation, Methodology, Writing – original draft, Project administration

## Author ORCIDs

Pierluigi Scerbo ⬭ https://orcid.org/0000-0002-3360-4367
David Bensimon ⬭ https://orcid.org/0000-0003-1971-9907

## Ethics

The study was performed in strict accordance with CNRS rules for use of laboratory animals and following the directives of the fish facility of the IBENS (agreement n°D750532; authorization APAFIS #47641-2024022014082203 v6).

Reviewer #2 (Public review): https://doi.org/10.7554/eLife.97650.3.sa1
Reviewer #3 (Public review): https://doi.org/10.7554/eLife.97650.3.sa2
Author response https://doi.org/10.7554/eLife.97650.3.sa3

---

# Additional files

## Supplementary files

MDAR checklist

## Data availability

RTqPCR data files for Figure 1—figure supplement 5 and 6 have been provided in *Figure 1—figure supplement 5—source data 1*.

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
