## [Editor Report · eLife Assessment]

This **important** study employs an optogenetics approach aimed at activating oncogene (KRASG12V) expression in a single somatic cell, with a focus on following the progression of activated cell to examine tumourigenesis probabilities under altered tissue environments. Although the description of the methodologies applied is **incomplete**, the authors propose a mechanism whereby reactivation of re-programming factors correlates with the increased likelihood of a mutant cell undergoing malignant transformation. This work will be of interest to developmental and cancer biologists, especially in relation to the genetic tools described.

---

## [Referee Report · Reviewer #2 (Public review)]

Summary:

In the work by Scerbo et al, the authors aim to better understand the open question of what factors constrain cells that are genetically predisposed to form cancer (e.g. those with a potentially cancer-causing mutation like activated Ras) to only infrequently undergo this malignant transformation, with a focus on the influence of embryonic or pluripotency factors (e.g. VENTX/NANOG). Using genetically defined zebrafish models, the authors can inducibly express the KRASG12V oncogene using a combination of Cre/Lox transgenes further controlled by optogenetically inducible Cre-activated (CreER fusion that becomes active with light-induced uncaging of a tamoxifen-analogue in a targeted region of the zebrafish embryo). They further show that transient expression and activation of a pluripotency factor (e.g. Ventx fused to a GR receptor that is activated with addition of dexamethasone) must occur in the model in order for overgrowth of cells to occur. This paper describes a genetically tractable and modifiable system for studying the requirements for inducing cellular hyperplasia in a whole organism by combining overexpression of canonical genetic drivers of cancer (like Ras) with epigenetic modifiers (like specific transcription factors), which could be used to study an array of combinations and temporal relationships of these cancer drivers/modifiers.

Strengths:

The combination of Cre/lox inducible gene expression with potentially localized optogenetic induction (CreER and uncaging of tamoxifen analogues) of recombination as well as inducible activation of a transcription factor expressed via mRNA injection (GR-fusion to the TF and dex induction) offers a flexible system for manipulating cell growth, identity, and transcriptional programs. With this system, the authors establish that Ras activation and at least transient Ventx overexpression are together required to induce a hyperproliferative phenotype in zebrafish tissues.

The ability to live image embryos over the course of days with inducible fluorophores indicating recombination events and transgene overexpression offers a tractable in vivo system for studying hyperplastic cells in the context of a whole organism.

The transplant experiments demonstrate the ability of the induced hyperplastic cells to grow upon transfer to new host.

Weaknesses:

There is minimal quantitation of key aspects of the system, most critically in the efficiency of activation of the Ras-TFP fusion (Fig 1) in, purportedly, a single cell. The authors note "On average the oncogene is then activated in a single cell, identified within ~1h by the blue fluorescence of its nuclear marker but no additional quantitative information is provided. For a system that is aimed at "a statistically relevant single-cell

tracking and characterization of the early stages of tumorigenesis", such information seems essential.

The authors indicate that a single cell is "initiated" (Fig 2) using the laser optogenetic technique, but without definitive genetic lineage tracing, it is not possible to conclude that cells expressing TFP distant from the target site near the ear are daughter cells of the claimed single "initiated" cell. A plausible alternative explanation is (1) that the optogenetic targeting is more diffuse (i.e. some of the light of the appropriate wavelength hits other cells nearby due to reflection/diffraction), so these adjacent cells are additional independent "initiated" cells or (2) that the uncaged tamoxifen analogue can diffuse to nearby cells and allow for CreER activation and recombination. In Fig 2B, the claim is made that "the activated cell has divided, giving rise to two cells" - unless continuously imaged or genetically traced, this is unproven. In addition, it appears that Figures S3 and S4 are showing that hyperplasica can arise in many different tissues (including intestine, pancreas, and liver, S4C) with broad Ras + Ventx activation (while unclear from the text, it appears these embryos were broadly activated and were not "single cell activated using the set-up in Fig 1E? This should be clarified in the manuscript). In Fig S7 where single cell activation and potential metastasis is discussed, similar gut tissues have TFP+ cells that are called metastatic, but this seems consistent with the possibility that multiple independent sites of initiation are occurring even when focal activation is attempted.

Although the hyperplastic cells are transplantable (Fig 4), the use of the term "cells of origin of cancer" or metastatic cells should be viewed with care in the experiments showing TFP+ cells (Fig 1, 2, 3) in embryos with targeted activation for the reasons noted above.

Comments on latest version:

The authors have clarified and strengthened a number of important conclusions/claims.

In Figure 4, the requirement for both kRas and VentX activation for successful transplant and survival of transplanted activated cells does indeed support the need for both MAPK activation and the reprogramming factor. A limitation remains that, as in a tail vein injection in a mouse model, this may be a better measure of the ability of disbursed cells to survive in the embryo, and not "native" metastatic behavior as cells may just lodge in ectopic sites, and survive, but not exhibit complete metastatic potential. Still, these are interesting and important results about the combination effects of an oncogene and a reprogramming factor.

Further, the addition of Fig 2A and additional explanation in the text on the specificity of the light-induced activation of the Ras and/or VentX supports that transgene induction is indeed limited to one or a few cells. We agree that visual tracking of daughter cells over days is technically challenging and will be a revealing and exciting potential addition in the future.

---

## [Referee Report · Reviewer #3 (Public review)]

Summary:

This study employs an optogenetics approach aimed at activating oncogene (KRASG12V) expression in a single somatic cell, with a focus on following the progression of activated cell to examine tumourigenesis probabilities under altered tissue environments. The research explores the role of stemness factors (VENTX/NANOG/OCT4) in facilitating oncogenic RAS (KRASG12V)-driven malignant transformations. Although the evidence provided is incomplete, the authors propose an important mechanism whereby reactivation of re-programming factors correlates with the increased likelihood of a mutant cell undergoing malignant transformation.

Strengths:

· Innovative Use of Optogenetics: The application of optogenetics for precise activation of KRAS in a single cell is valuable to the field of cancer biology, offering an opportunity to uncover insight into cellular responses to oncogenic mutations.

· Important Observations: The findings concerning stemness factors' role in promoting oncogenic transformation are important, contributing data to the field of cancer biology.

Weaknesses:

Lack of Methodological Clarity: The manuscript lacks detailed descriptions of methodologies, making it difficult to fully evaluate the experimental design and reproducibility, rendering incomplete evidence to support the conclusion. Improving methodological transparency and data presentation will crucially strengthen the paper's contributions to understanding the complex processes of tumorigenesis.

Sub-optimal Data Presentation and Quality:

The resolution of images through-out the manuscript are too low. Images presented in Figure 2 and Figure 4 are of very low resolution. It is very hard to distinguish individual cells and in which tissue they might reside.

Lack of quantitative data and control condition data obtained from images of higher magnification limits the ability to robustly support the conclusions.

Here are some details:

· Tissue specificity of the cells express KRASG12V oncogene: In this study, the ubiquitin promoter was used to drive oncogenic KRASG12V expression. Despite this, the authors claim to activate KRAS in a single brain cell based on their localized photo-activation strategy. However, upon reviewing the methods section, the description was provided that 'Localized uncaging was performed by illumination for 7 minutes on a Nikon Ti microscope equipped with a light source peaking at 405 nm, Figure 1. The size of the uncaging region was controlled by an iris that defines a circular illumination with a diameter of approximately 80 μm.' It is surprising that an epi-fluorescent microscope with an illumination diameter of around 80μm can induce activation in a single brain cell beneath skin tissue. Additionally, given that the half-life for mTFP maturation is around 60 minutes, it is likely that more cells from a variety of different lineages could be activated, but the fluorescence would not be visible until more than 1-hour post-illumination. Authors might want to provide more evidence to support their claim on the single cell KRAS activation.

· Stability of cCYC: The manuscript does not provide information on the half-life and stability of cCYC. Understanding these properties is crucial for evaluating the system's reliability and the likelihood of leakiness, which could significantly influence the study's outcomes.

· Metastatic Dissemination claim: Typically, metastatic cancer cells migrate to and proliferate within specific niches that are conducive to outgrowth, such as the caudal hematopoietic tissue (CHT) or liver. In Figure 3 A, an image showing the presence of mTFP expressing cells in both the head and tail regions of the larva, with additional positive dots located at the fin fold. This is interpreted as "metastasis" by the authors. However, the absence of supportive cellular compartment within the fin-fold tissue makes the presence of mTFP-positive metastatic cells there particularly puzzling. This distribution raises concerns about the spatial specificity of the optogenetic activation protocol.

The unexpected locations of these signals suggest potential ectopic activation of the KRAS oncogene, which could be occurring alongside or instead of targeted activation. This issue is critical as it could affect the interpretation of whether the observed mTFP signal expansion over time is due to actual cell proliferation and infiltration, or merely a result of ectopic RAS transgene activation.

· Image Resolution Concerns: The cells depicted in Figure 3C β, which appear to be near the surface of the yolk sac and not within the digestive system as suggested in the MS, underscore the necessity for higher-resolution imaging. Without clearer images, it is challenging to ascertain the exact locations and states of these cells, thus complicating the assessment of experimental results.

· The cell transplantation experiment is lacking protocol details: The manuscript does not adequately describe the experimental protocols used for cell transplantation, particularly concerning the origin and selection of cells used for injection into individual larvae. This omission makes it difficult to evaluate the reliability and reproducibility of the results. Such as the source of transplanted cells:

• If the cells are derived from hyperplastic growths in larvae where RAS and VX (presumably VENTX) were locally activated, the manuscript fails to mention any use of fluorescence-activated cell sorting (FACS) to enrich mTFP-positive cells. Such a method would be crucial for ensuring the specificity of the cells being studied and the validity of the results.

• If the cells are obtained from whole larvae with induced RAS + VX expression, it is notable and somewhat surprising that the larvae survived up to six days post-induction (6dpi) before cells were harvested for transplantation. This survival rate and the subsequent ability to obtain single cell suspensions raise questions about the heterogeneity of the RAS + VX expressing cells that transplanted.

· Unclear Experimental Conditions in Figure S3B: The images in Figure S3B lack crucial details about the experimental conditions. It is not specified whether the activation of KRAS was targeted to specific cells or involved whole-body exposure. This information is essential for interpreting the scope and implications of the results accurately.

· Contrasting Data in Figure S3C compared to literatures: The graph in Figure S3C indicates that KRAS or KRAS + DEX induction did not result in any form of hyperplastic growth. This observation starkly contrasts with previous literature where oncogenic KRAS expression in zebrafish led to significant hyper-proliferation and abnormal growth, as evidenced by studies such as those published in and Neoplasia (2018), DOI: 10.1016/j.neo.2018.10.002; Molecular Cancer (2015), DOI: 10.1186/s12943-015-0288-2; Disease Models & Mechanisms (2014) DOI: 10.1242/dmm.007831. The lack of expected hyperplasia raises questions about the experimental setup or the specific conditions under which KRAS was expressed. The authors should provide detailed descriptions of the conditions under which the experiments were conducted in Figure S3B and clarifying the reasons for the discrepancies observed in Figure S3C are crucial. The authors should discuss potential reasons for the deviation from previous reports.

Further comments:

Throughout the study, KRAS-activated cell expansion and metastasis are two key phenotypes discussed that Ventx is promoting. However, the authors did not perform any experiments to directly show that KRAS+ cells proliferate only in Ventx-activated conditions. The authors also did not show any morphological features or time-lapse videos demonstrating that KRAS+ cells are motile, even though zebrafish is an excellent model for in vivo live imaging. This seems to be a missed opportunity for providing convincing evidence to support the authors' conclusions.

There were minimal experimental details provided for the qPCR data presented in the supplementary figures S5 and S6, therefore, it is hard to evaluate results obtained.

---

## [Author Response]

The following is the authors’ response to the original reviews.

First, we thank the reviewers for a thorough reading of our paper and some useful comments. A recurrent remark of the reviewers concerns the appearance of kRas-expressing cells (labelled by a nuclear blue fluorescent marker) which we attribute to the progeny of the initially induced cell. The reviewers suggest that these cells may have been obtained through activation of the Cre-recombinase in other cells by cyclofen released from light scattering, via diffusion, leakiness, etc. These remarks are perfectly reasonable from people not familiar with the cyclofen uncaging approach that we are using, but are unwarranted as we shall show below.

We have been using cyclofen uncaging with subsequent activation of a Cre-recombinase (or some other proteins) since 2010 see ref.34, Sinha et al., Zebrafish 7, 199-204 (2010) and our 2018 review (ref.35, Zhang et al., ChemBioChem 19,1-8 (2018)). In our experiments, the embryos are incubated in the dark in 6µM caged cyclofen (cCyc) and washed in E3 medium (and transferred to a new medium with no cCyc). In these conditions, over many years we never observed activation of the recombinase, i.e. the appearance of the associated fluorescent label in cells of embryos grown in E3 medium. Hence leakiness can be ruled out (in presence of cCyc or in its absence).

Following transfer of the embryos to new E3 medium we illuminate the embryos locally with light at 405nm. In these conditions, cCyc is only partially uncaged and results in activation of Cre-recombinase in only a few cells (1,2, 3, …) within the illuminated region only, namely in the appearance of the kRas-associated nuclear blue fluorescent label in usually one cell (and sometimes in a few more). Data and statistics are now incorporated in the revised manuscript, see Fig.2A and S7. In absence of activation of a reprogramming factor these fluorescently labelled cells disappear within a few days (either via shut-down of their promotor, apoptosis or some other mechanism). The crucial point here is that we see less and not more kRas expressing cells (i.e. with nuclear blue fluorescence) in absence of VentX activation. This observation rules out activation of Cre-recombinase in other cells days after illumination due to leakiness, cyclofen released by light or diffusing from the illumination spot.

To observe many more fluorescent cells days after activation of the initial cell, one needs to transiently activate VentX-GR by overnight incubation in dexamethasone (DEX). Injecting the embryos at 1-cell stage with VentX-GR only or incubating them in DEX (without injection of VentX-GR) does not result in the appearance of more blue fluorescent cells. Following activation of VentX-GR, the fluorescent cells observed a couple of days after initiation are visualized in E3 medium (i.e. in absence of cyclofen) and are localized to the vicinity of the otic vesicle (the region where the initial cell was activated). In the revised manuscript we show images of these fluorescent cells taken a few days apart in the same embryo in which a single cell was initially activated (Fig.S8). Hence, we attribute these cells to the progeny of the activated cell. Obviously, single cell tracking via time-lapse microscopy would definitely nail down this issue and provide fascinating insight into the initial stages of tumor growth. Unfortunately, immobilization of embryos in the usual medium (e.g. MS222, tricaine) over 5-6 days to track the division and motion of single cells is not possible. We are considering some other possibilities (immobilization in bungarotoxin or via photo-activation of anionic channels), but these challenging experiments are for a future paper.

**Reviewer #1 (Public Review):**
The authors then performed allotransplantations of allegedly single fluorescent TICs in recipient larvae and found a large number of fluorescent cells in distant locations, claiming that these cells have all originated from the single transplanted TIC and migrated away. The number of fluorescent cells showed in the recipient larve just after two days is not compatible with a normal cell cycle length and more likely represents the progeny of more than one transplanted cell.

As mentioned in the manuscript, we measure the density of cells/nl and inject in the yolk of 2dpf Nacre embryos a volume equivalent to about 1 cell, following published protocols (S.Nicoli and M.Presta, Nat.Prot. 2,2918 (2007)). We further image the injected cell(s) by fluorescence microscopy immediately following injection, as shown in Fig.4A and Fig.S8B. We might miss a few cells but not many. With a typical cell cycle of ~10h the images of tumors in larvae at 3dpt (and not 2dpt) correspond to ~100 cells. In any case the purpose of this experiment was to show that the progeny of the initial induced cell is capable of developing into a tumor in a naïve fish, which is the operational definition of cancer that we adopted here.

The ability to migrate from the injection site should be documented by time-lapse microscopy.

As stated above our purpose here is not to study tumor formation from transplanted cell(s) but to use that assay as an operational test of cancer. Besides as mentioned earlier single cell tracking in larvea over 3-4dpt is not a trivial task.

Then, the authors conclude that "By allowing for specific and reproducible single cell malignant transformation in vivo, their optogenetic approach opens the way for a quantitative study of the initial stages of cancer at the single cell level". However, the evidence for these claims are weak and further characterization should be performed to:(1) Show that they are actually activating the oncogene in a single cell (the magnification is too low and it is difficult to distinguish a single nucleus, labelling of the cell membrane may help to demonstrate that they are effectively activating the oncogene in, or transplanting, a single cell)

In the revised manuscript we provide larger magnification of the initial induced cell and show examples of oncogene activation in more than one cell.

(2) The expression of the genes used as markers of tumorigenesis is performed in whole larvae, with only a few transformed cells in them. Changes should be confirmed in FACS sorted fluorescent cells

When the oncogene is activated in a whole larvae all cells are fluorescent and thus FACS is of no use for cell sorting. Sorting could be done in larvae where single cells are activated , but then the efficiency of FACS is not good enough to isolate the few fluorescent cells among the many more non-fluorescent ones. We agree that the expression change of the genes used as markers of tumorigenesis is an underestimate of their true change, but our goal at this time is not to precisely measure the change in expression level, but to show that the pattern of change was different from the controls and corresponded to what is expected in tumorigenesis.

(3) The histology of the so called "tumor masses" is not showing malignant transformation, but at the most just hyperplasia.

The histology of the hyperplasic tissues show cellular proliferation with a higher density of nuclear material which is characteristic of tumors, Fig.S4C. Besides the increased expression of pERK in these tissues, Fig.S4A,B is also a hallmark of cancer.

In the brain, the sections are not perfectly symmetrical and the increase of cellularity on one side of the optic tectum is compatible with this asymmetry.

The expected T-shape formed by the sections of the tegmentum and hypothalamus are compatible with the symmetric sections shown in Fg.2D. The asymmetry in the optic tectum is a result of the hyperplasic growth.

(4) The number of fluorescent cells found dispersed in the larvae transplanted with one single TIC after 48 hours will require a very fast cell cycle to generate over 50 cells. Do we have an idea of the cell cycle features of the transplanted TICs?

As answered above, the transplanted larvae are shown at 3dpt. With a cell cycle of about 10h, a single cell can give rise to about 100 cells in that time lapse.

**Reviewer #2 (Public Review):**
Summary:This paper describes a genetically tractable and modifiable system …which could be used to study an array of combinations and temporal relationships of these cancer drivers/modifiers.

We thank this referee for its positive comments. We would also like to point out that our approach provides for the first quantitative means to estimate the probability of tumorigenesis from a single cell, an estimate which is crucial in any assessment of cancer malignancy and the effectiveness of prophylactics.

Weaknesses:There is minimal quantitation of … the efficiency of activation of the Ras-TFP fusion (Fig 1) in, purportedly, a single cell. …, such information seems essential.

We have added more images of induction of a single (or a few cells) and a plot where the probability of RAS activation in one or a few cells is specified.

The authors indicate that a single cell is "initiated" (Fig 2) using the laser optogenetic technique, but without definitive genetic lineage tracing, it is not possible to conclude that cells expressing TFP distant from the target site near the ear are daughter cells of the claimed single "initiated" cell. A plausible alternative explanation is (1) that the optogenetic targeting is more diffuse (i.e. some of the light of the appropriate wavelength hits other cells nearby due to reflection/diffraction), so these adjacent cells are additional independent "initiated" cells or (2) that the uncaged tamoxifen analogue can diffuse to nearby cells and allow for CreER activation and recombination.

We have addressed this point in our general comments to the reviewers’ remarks. The possibilities mentioned by this reviewer would result in cells expressing TFP in absence of VentX activation, which is NOT the case. Cells expressing TFP away from the initial site are observed DAYS after activation of the oncogene (and TFP) in a single cell and ONLY upon activation of VentX.

In Fig 2B, the claim is made that "the activated cell has divided, giving rise to two cells" - unless continuously imaged or genetically traced, this is unproven.

We have addressed this remark previously. Tracking of larvae over many days is not possible with the usual protocol using tricaine to immobilize the larvae. Nonetheless, in the revised version we present images of an embryo imaged at various times post activation (1hpi, 3dpi, 7dpi) where proliferation and metastasis of the cells can be observed. We are pursuing other alternatives for time-lapse microscopy over many days, since besides convincing the sceptics, a single cell tracking experiment (possibly coupled with in-situ spatial transcriptomics) will shed a new and fascinating light on the initial stages of tumor growth.

In addition, it appears that Figures S3 and S4 are showing that hyperplasia can arise in many different tissues (including intestine, pancreas, and liver, S4C with broad Ras + Ventx activation …. This should be clarified in the manuscript).

This is true and has been clarified in the new version.

In Fig S7 where single cell activation and potential metastasis is discussed, similar gut tissues have TFP+ cells that are called metastatic, but this seems consistent with the possibility that multiple independent sites of initiation are occurring even when focal activation is attempted.

As mentioned previously this is ruled out by the fact that these cells are observed days after cyclofen uncaging (and TFP activation) and IF AND ONLY IF VentX was activated during the first dpi.

Although the hyperplastic cells are transplantable (Fig 4), the use of the term "cells of origin of cancer" or metastatic cells should be viewed with care in the experiments showing TFP+ cells (Fig 1, 2, 3) in embryos with targeted activation for the reasons noted above.

The purpose of this transplantation experiment was to show that cell in which both kRas and VentX have been activated possess the capacity to metastasize and develop a tumor mass when transplanted in a naïve zebrafish. This - to the best of our knowledge - is the operational definition of a malignant tumor. Notice also that transplantation of kRAS only activated cells (i.e. without subsequent activation of VentX) does NOT yield tumors, rather the transplanted cell disappears after a few days, see Fig.S10.

**Reviewer #3 (Public Review):**
Summary:This study employs an optogenetics approach … to examine tumorigenesis probabilities under altered tissue environments.

We thank this reviewer for this remark, since we believe that the probability to assess the probability of tumorigenesis from a single cell is probably the most significant contribution of this work.

Weaknesses:Lack of Methodological Clarity: The manuscript lacks detailed descriptions of methodologies,

We have included additional detail of our methodology and statistical analyses in the revised manuscript.

Sub-optimal Data Presentation and Quality:Lack of quantitative data and control condition data obtained from images of higher magnification limits the ability to robustly support the conclusions.

We have included more images at higher magnification and quantitative data to support the main report of targeted single cell induction.

Here are some details:Authors might want to provide more evidence to support their claim on the single cell KRAS activation.

More images and a data on activation of single or few cells in the illumination field are provided as well as statistical analysis of cell induction.

Stability of cCYC: The manuscript does not provide information on the half-life and stability of cCYC. Understanding these properties is crucial for evaluating the system's reliability and the likelihood of leakiness, which could significantly influence the study's outcomes.

We have been using the cCyc system for about 14 years. We refer the reader to our previous papers and reviews on this methodology. Briefly, cCyc is stable when not illuminated with light around 375nm. Typically, we incubate our embryos in the dark for about 1h before washing, transferring them into E3 medium and illuminating them. Assessing the leakiness of the system is easy as expression of a fluorescent marker is permanently turned on. We have observed none in the conditions of our experiment or in previous works.

Metastatic Dissemination claim: However, the absence of a supportive cellular compartment within the fin-fold tissue makes the presence of mTFP-positive metastatic cells there particularly puzzling. This distribution raises concerns about the spatial specificity of the optogenetic activation protocol … The unexpected locations of these signals suggest potential ectopic activation of the KRAS oncogene,

We have addressed this remark in the introduction and above. Specifically, metastatic and proliferative mTFP-positive cells are observed IF AND ONLY IF VentX is also activated concomitant with activation of kRAS in a single cell. No proliferative cells are observed in absence of VentX activation, or in presence of VentX or Dex alone, or if kRAS has not been activated by cyclofen uncaging.

Image Resolution Concerns: The cells depicted in Figure 3C β, which appear to be near the surface of the yolk sac and not within the digestive system as suggested in the MS, underscore the necessity for higher-resolution imaging. Without clearer images, it is challenging to ascertain the exact locations and states of these cells, thus complicating the assessment of experimental results.

Better images are provided in the revised version.

The cell transplantation experiment is lacking protocol details:

Details are provided. We have followed regular protocols for transplantation: S.Nicoli and M.Presta, Nat.Prot. 2,2918 (2007).

If the cells are obtained from whole larvae with induced RAS + VX expression, it is notable and somewhat surprising that the larvae survived up to six days post-induction (6dpi) before cells were harvested for transplantation. This survival rate and the subsequent ability to obtain single cell suspensions raise questions about the heterogeneity of the RAS + VX expressing cells that transplanted.

From Fig.S4D, about 50% of the embryos survive at 6dpi. Though an interesting question by itself we have not (yet) addressed the important issue of the heterogeneity of the outgrowth obtained from a single cell. Our purpose here was just to show that cells in which both kRAS and VentX have been activated possess the capacity to metastasize and develop a tumor mass when transplanted in a naïve zebrafish. This - to the best of our knowledge - is the operational definition of a malignant tumor.

Unclear Experimental Conditions in Figure S3B: …It is not specified whether the activation of KRAS was targeted to specific cells or involved whole-body exposure.

This was whole body (global) illumination and is specified in the revised version.

Contrasting Data in Figure S3C compared to literature: The graph in Figure S3C indicates that KRAS or KRAS + DEX induction did not result in any form of hyperplastic growth. The authors should provide detailed descriptions of the conditions under which the experiments were conducted in Figure S3B and clarifying the reasons for the discrepancies observed in Figure S3C are crucial. The authors should discuss potential reasons for the deviation from previous reports.

This discrepancy is discussed in the revised version. First the previous reports consider the development of tumors within 3-4 weeks which we have not studied in detail. Second, the expression of the oncogene in these reports might be stronger than in ours. Third, the stochastic and random appearance of tumors in these reports suggest that some other mechanism (transient stress-induced reprogramming?) might have activated the oncogene in the initial cell.

Further comments:Throughout the study, KRAS-activated cell expansion and metastasis are two key phenotypes discussed that Ventx is promoting. However, the authors did not perform any experiments to directly show that KRAS+ cells proliferate only in Ventx-activated conditions.

Yes, we did. See Fig. S1 and compare with Fig.S3B, or Fig.S10A in comparison with Fig.2A,B.

The authors also did not show any morphological features or time-lapse videos demonstrating that KRAS+ cells are motile, even though zebrafish is an excellent model for in vivo live imaging. This seems to be a missed opportunity for providing convincing evidence to support the authors' conclusions.

Performing time-lapse microscopy on larvae over many (4-5) days is not possible with the regular tricaine protocol for immobilization. We are definitely planning such experiments, but they will require some other protocol, perhaps using bungarotoxin or some optogenetic inhibitory channels.

There were minimal experimental details provided for the qPCR data presented in the supplementary figures S5 and S6, therefore, it is hard to evaluate result obtained.

More details are given in the revised version.

**Recommendations for the authors:**

**Reviewer #1 (Recommendations For The Authors):**
Abstract: what is the definition of tumors that they are using? I never heard of a full-blown tumor that develops in less than 6 days from a single cell!

This is indeed surprising! We are using an operational definition of a tumor: if cells from an hyperplasic tissue can metastasize and outgrow when transplanted in a naïve zebrafish, then it is a tumor.

Introduction: The claim that this is the first report of the induction of oncogene expression in a single cell in zebrafish is wrong as there are other reports (PMID: 27810924, PMID: 30061297)

These other approaches are invasive (electroporation and transplantation). We have added non-invasive in the revised version.

Figure 2: The quality of these images is too low to visualize the infiltration that they talk about, the sections are not perfectly coronal and the asymmetric distribution of cells may be confused with an infiltration.

We have addressed this question above.

Results, page 5: how do we know that these are metastatic cells? there could have been spurious activation in other locations, you need to prove that these cells moved from one place to the other and that they are of the same cell type as the primary tumor

We have addressed this question extensively in the introduction and in our answers to the reviewers. We have also added a figure showing cell proliferation in the same embryos at various time post induction. Time-lapse microscopy studies of tumor initiation and growth over many days are planned, but will be the subject of an other paper.

Figure 3: not clear why they did not use anaesthetic or mounting media to take pictures of the transplanted fish

We tried to minimally stress the larvae that are already in a perilous condition…

Results, page 6: Not clear why the authors used KRAS v12 as an oncogene and uncaged its expression in the brain, as KRAS is not a common oncogene for brain tumors.

There are reports of kRASG12V tumors in zebrafish brain (doi: 10.1186/s12943-015-0288-2)

It is not clear what is the mechanism of Ventx -driven oncogenesis? What changes in gene expression, cell function etc are induced by Ventx in the cells that express KRASv12? The qPCR analysis performed is done on whole larvae and an analysis on single TICs and their progeny should be done following FACS sorting of fluorescent cells.

FACS sorting of a single TIC (and its progeny) among many thousand cells in the embryo is not possible. The analysis on whole larvae provides an underestimate of the changes in gene expression following activation of kRAS and VentX. We are looking for spatial transcriptomics as a better approach of the changes in gene expression induced in single TICs and their progeny, but that is beyond the scope of this paper.

Nuclear staining is necessary to make sure that only 1 cell was transplanted. How is it possible that we get more than 50 cells from a single transplanted cell in less than 48 hours? What is the length of the cell cycle of these transformed cells?

Nuclear staining is not necessary as the transplanted cell is fluorescent. Thus we can see how many cells are transplanted. With a cell-cycle of about 10h in 3dpt, a single cell will have generated as many as 100 cells.

**Reviewer #2 (Recommendations For The Authors):**
Minor grammatical change - hyperplasic more commonly called hyperplastic.
**Reviewer #3 (Recommendations For The Authors):**
Provide Detailed Methodologies: Clearly describe all experimental protocols used, particularly those for cell transplantation and photo-activation techniques. Detailed protocols will aid in replicating your findings and enhancing the manuscript's credibility.

Done.

Provide High-Resolution Imaging data: To substantiate the claims about cell location and behaviour, provide high-resolution images where individual cells and their specific tissue contexts are clearly visible.

Greater magnification images provided.

Quantitative Data: Incorporate quantitative analyses to strengthen the findings, particularly in experiments where cell proliferation and activation are key outcomes.

Done.

Verify Single Cell Activation: Offer additional evidence or experimental validation to support the claim that KRASG12V activation is confined to single cells, considering the limitations mentioned about the photo-activation setup.

Discussion, figures and statistical analysis added in manuscript.

Discuss Stability and Leakage of cCYC: Provide data on the stability and half-life of cCYC to assess the likelihood of system leakiness, which could influence the interpretation of your results.

Reference to our previous papers and reviews added.

Clarify Metastatic Claims: Discuss the unexpected presence of mTFP-positive cells in nontraditional metastatic sites, like the fin fold, and consider additional experiments to verify whether these are cases of ectopic activation or true metastasis.

Discussion added in manuscript

Utilize time-lapse live imaging to visually document the motility and behaviour of KRAS+ cells over time, leveraging the strengths of the zebrafish model.

Definitely interesting, but non trivial to conduct over many days and subject for a future paper.

Address Discrepancies in KRAS Activation Effects from literature: Specifically, discuss why your findings on KRAS-induced hyperplasia differ from existing literature. Consider whether experimental conditions or KRAS expression levels might have contributed to these differences.

Discussion added in revised version